# Outpatient Antibiotic Use and Costs in Adults: A Nationwide Register-Based Study in Finland 2008–2019

**DOI:** 10.3390/antibiotics11111453

**Published:** 2022-10-22

**Authors:** Elisa Pyörälä, Kati Sepponen, Anneli Lauhio, Leena Saastamoinen

**Affiliations:** 1School of Pharmacy, Faculty of Health Sciences, University of Eastern Finland, 70211 Kuopio, Finland; 2Faculty of Medicine, University of Helsinki, 00014 Helsinki, Finland; 3Finnish Medicines Agency Fimea, 00300 Helsinki, Finland; 4The Social Insurance Institution, 00250 Helsinki, Finland

**Keywords:** antibiotics, antibacterial, antibacterial drug, antibiotic use, antimicrobial resistance, antibiotic resistance, cost analysis, Gram-negative threat, *E. coli* resistance, new innovations

## Abstract

The objective of this study was to describe the prevalence of outpatient use and costs for systemic antibacterials by age and sex among adults in Finland from 2008–2019. Data from the Finnish statistical database Kelasto, containing information concerning all reimbursed medicines for 18+-year-olds during 2008–2019, were analyzed. In addition to the decreased (26%) use of systemic antibiotics, decreased use was observed in all antibiotic categories, notably including several wide-spectrum antibiotics. The use of quinolones decreased by 49% and of tetracyclines by 39%. The 10 most frequently used antibiotics covered 89% of all adult antibiotic prescriptions. Antibiotic use also decreased in every age group during the study period. Although the overall yearly costs of outpatient antibiotics during the 10-year study period decreased from EUR 36.4 million to EUR 30.7 million, the cost per prescription increased slightly. In conclusion, according to the findings of this study, concerning adults and the results of our previous study concerning children and adolescents (2008–2016), there has been a decreasing trend of outpatient antibacterial use among the whole Finnish outpatient population over the duration of nearly one decade. However, during the same time period, there has been a specific increasing trend for the Gram-negative AMR threat regarding *E. coli* resistance. Therefore, based on our important findings in Finland, methods other than the restriction of antibiotic use, such as new anti-infective innovations, including antibacterials, are needed as soon as possible to tackle this major global health threat—a silent pandemic.

## 1. Introduction

Antimicrobial resistance (AMR) is a major health threat around the world [1]. AMR causes increased mortality, complications, longer hospital stays, and higher health expenses due to antibacterial drugs no longer being effective for infectious diseases that used to be easily treatable. The WHO and EU have launched action plans on fighting antimicrobial resistance [2,3]. Finland has also launched a national action plan (NAP) on AMR with the goal of reducing the use of antibacterial drugs [4]. In a recent study of the global AMR burden, the Gram-negative threat was found to be the leading cause of death, with the Gram-negative bacteria *Escherichia coli* being the leading pathogen for death associated with resistance [1]. The fact that no new antibiotics nor class of antibacterial drugs with a new mechanism of action has come to the market for decades [5] further emphasizes the importance of this major health threat.

In Finland, like in many other countries, *Escherichia coli* and *Klebsiella pneumonia* ESBL findings were published in the early 2000s [6], with an increasing trend during the ten-year period of 2006–2015 reflecting the increasing Gram-negative threat, especially in the treatment of urinary tract infection [7,8]. However, in our previous nationwide register-based study from 2008–2016, we found a clear decreasing trend of outpatient antibiotic use among children and adolescents in Finland [9]. There is a common opinion that increasing AMR is associated with increasing use of antibiotics [10]. Therefore, there was also a need to continue to study the use of antibiotics among the adult population during the same time period in Finland. To the best of our knowledge, there are several studies on the topic from other countries, but mostly covering a shorter time period, using a smaller sample size [11,12,13,14,15,16,17,18,19,20,21], or using statistics without considering patients’ age or sex [22,23]. The present study was carried out before the COVID-19 pandemic, which had drastic effects [24] on the use of antibiotics due to several infection control orders in Finnish society. The short-term effect of the COVID-19 pandemic on antibiotic use in Finland has been studied elsewhere [25]. With this background, we set up the present nationwide work to study the use of antibiotics by age and sex among the total adult outpatient Finnish population from 2008–2019, and also the cost of antibiotics.

## 2. Results

### 2.1. Antibiotic Use

The overall antibiotic prescription rate use decreased by 26% from 582 prescriptions per 1000 adults in 2008 to 428 prescriptions per 1000 adults in 2019 (Figure 1). The antibiotic prescription rate was highest in 2011 (614 prescriptions per 1000 adults), after which the rate decreased steadily. The average rate during the entire study period from 2008–2019 was 534 prescriptions per 1000 adults per year. The proportion of adults who filled at least one antibiotic prescription decreased from 33% in 2008 to 26% in 2019.

During the entire research period (2008–2019), antibiotic use was substantially higher in females compared to males, but the decrease was similar in both sexes (Figure 1). The decrease was 26% in females (from 700 to 517 prescriptions per 1000 adults per year) and 26% in males (from 457 to 336 prescriptions per 1000 adults per year). The proportion of females who received at least one antibiotic prescription decreased from 38% in 2008 to 30% in 2019, and the decrease was also similar in men: from 28% to 21%.

The most used antibiotic groups during the research period were beta-lactam antibacterials, penicillins (J01C) (195 prescriptions per 1000 adults in 2019), other beta-lactam antibacterials (J01D) (104 prescriptions per 1000 adults), and tetracyclines (J01A) (52 prescriptions per 1000 adults) (Table 1). The most common subgroups of the penicillins were penicillins with an extended spectrum, beta-lactamase-sensitive penicillins, and combinations of penicillins. Of the other beta-lactam antibacterial prescriptions, 98% were first-generation cephalosporins, primarily cefalexin. The decrease in the antibiotic prescription rate was seen in all antibiotic categories presented in Table 1. The decrease was greatest in the group of macrolides, lincosamides and streptogramins (J01F, −60%), and quinolone antibacterials (J01M, −49%).

The 10 most frequently used individual antibiotics covered 89% of all antibiotic prescriptions from 2008–2019. Cefalexin was the most common antibiotic throughout the research period (Figure 2), covering 24% of all antibiotic prescriptions in 2019, followed by amoxicillin, except for the years 2008, 2011, and 2012, when doxycycline was slightly more common than amoxicillin. Following on from this were doxycycline, phenoxymethylpenicillin, and pivmecillinam.

When comparing the antibiotic prescription rate between 2008 and 2019, a decrease was observed for 8 of the 10 most common antibiotics: the prescription rate of azithromycin decreased by 55%, doxycycline 39%, ciprofloxacin 36%, phenoxymethylpenicillin 32%, clindamycin 24%, cefalexin 18%, trimethoprim 8%, and amoxicillin/beta-lactamase inhibitor use 3%. However, a slight increase was seen in the use of doxycycline and azithromycin from 2010–2011.

Use of antibiotics decreased notably in adults in every age group during the research period (Figure 3). The greatest decrease in antibiotic use occurred in adults aged 30–39 years (37%), 40–49 years (32%) and 18–29 years (29%). The smallest percentage change was observed among adults aged ≥80 years (14%).

The antibiotic prescription rate was lowest in adults aged 18–29 years (357 prescriptions per 1000 adults in 2019) and highest in adults aged ≥80 years (731 prescriptions per 1000 adults in 2019). The prescription rate increased with age with relative stability until the age of 80 years, after which a steep increase in prescription rate was detected (Figure 3). The prescription rate of antibiotics used to treat urinary tract infections especially increased after the age of 80 years. In adults aged ≥80 years, over a third (36%) of all antibiotic prescriptions were pivmecillinam or trimethoprim prescriptions, whereas, in adults aged <80 years, only 8% of all antibiotic prescriptions were pivmecillinam or trimethoprim prescriptions.

The annual total costs of antibiotic use in adults decreased by 16% from EUR 36.4 million in 2008 to EUR 30.7 million in 2019, although the average cost per prescription increased (Figure 4). The highest total costs were seen in 2011 (EUR 39.9 million).

### 2.2. Antibiotic Costs

The costs per prescription increased by 8% from EUR 14.76 in 2008 to EUR 15.99 in 2019, being highest in 2016 (EUR 17.16) and lowest in 2013 (EUR 14.29) (Figure 4). The costs per prescription were higher in males than in females (EUR 17.27 vs. EUR 15.18 in 2019). Additionally, an age of ≥60 years was found to be associated with slightly higher costs per prescription (EUR 17.48 in adults aged ≥60 years vs. EUR 14.89 euros in adults aged <60 years in 2019).

The most used antibiotic cefalexin incurred the highest total costs: 22% of all antibiotic costs during the study period (Figure 5). The total costs of cefalexin decreased steeply between 2012 and 2013 and subsequently returned to an earlier level. The total costs of amoxicillin were the second highest (14% of all antibiotic costs), followed by amoxicillin/beta-lactamase inhibitor (8%), although the costs fluctuated strongly during the study period.

For the 10 most used antibiotics, the costs per prescription increased for cefalexin, amoxicillin, phenoxymethylpenicillin, doxycycline, clindamycin, and ciprofloxacin and decreased for the other substances. Of the 10 most used antibiotics, clindamycin was the highest-priced (EUR 32.60 per prescription in 2019). Of all antibiotics, phenoxymethylpenicillin (EUR 6.84 in 2019) and doxycycline (EUR 9.21 in 2019) had the lowest costs per prescription. The average cost per prescription of all the antibiotics in group J01 was EUR 16.27 in 2019.

## 3. Materials and Methods

### 3.1. Study Population

This study includes all the adults aged 18 years and over residing in Finland who purchased their antibiotics in community pharmacies.

### 3.2. Setting

In Finland, all oral antibiotics are prescription-only medicines. Antibiotics that are prescribed and granted reimbursement status by the Pharmaceuticals Pricing Board (Hila) are partially reimbursed to the patient under National Health Insurance, which covers all residents of Finland (5.53 million residents in 2019, 50.6% female) [24,26]. The Finnish medicine reimbursement system is administered by the Social Insurance Institution of Finland (Kela) [24]. Reimbursements (40% of the retail price) are granted after meeting the annual initial deductible of EUR 50.0 [24]. The majority of antibiotics have reimbursement status, but some antibiotics, such as nitrofurantoin and methenamine, are nonreimbursable.

### 3.3. Data Source

The research data was retrieved from the Finnish statistical database Kelasto administered by the Social Insurance Institution of Finland (Kela) [27]. The statistical database Kelasto is based on the Finnish Prescription Registry, which contains information on all reimbursed medicines dispensed to patients in Finnish pharmacies. In practice, most of the patients in nursing homes in Finland also purchase their medicines in pharmacies. Therefore, only hospital use (14% of antibiotic consumption measured in DDD/1000 inh/day in 2019) is excluded from the data [24]. The research data covered all residents of Finland aged ≥18 years who had purchased antibiotics defined as antibacterials for systemic use (J01) by the Anatomical Therapeutic Chemical (ATC) classification system and studied by ATC level 3 and level 5, i.e., substance level [28]. During the research period, no antibiotic prescriptions were observed in the ATC subgroups J01B (Amphenicols) or J01R (Combinations of antibacterials) since, in Finland, there are no medications in J01B or J01R that are approved for marketing. Number of reimbursement recipients, number of dispensed prescriptions, sex and age of the patient, name of the active drug ingredient, ATC code, reimbursement costs, and total costs were included in the data.

### 3.4. Analysis

The research data were analyzed using Microsoft Excel software (version 2008). The antibiotic prescription rate was expressed as the number of prescriptions per 1000 adults per year and the proportion of adults who filled at least one antibiotic prescription. The population figures were based on the population data, including the total Finnish population maintained by The Social Insurance Institution. The analyses were performed separately for both sexes and for all age groups (18–29, 30–39, 40–49, 50–59, 60–69, 70–79, and ≥80 years). In addition to the overall use of antibiotics, the prescription rate of the most common individual antibiotics and antibiotic groups was also examined. The 10 most frequently used antibiotics were chosen based on the total number of antibiotic prescriptions over the course of the entire study period.

### 3.5. Ethics

The research data include only aggregated register data and does not identify individuals. According to Finnish legislation, ethical approval was not required for this study.

## 4. Discussion

In this nationwide register-based study, we found a decreasing trend in the prescribing of antibacterial drugs during the study period (years 2008–2019) among the total adult outpatient population in Finland. The findings of the present study are similar to that of a previous study by Parviainen et al. [9] concerning children and adolescents from 2008–2016. Thus, it can be concluded, based on both studies, that we have shown a decreasing trend of antibiotic prescription rate among the whole Finnish outpatient population during a relatively long study period: about one decade.

Our result is conflicting when compared to the global antibiotic use during the 2000s, but Finland seems to follow the recent decreasing direction of the other high-income countries [29]. To control AMR, Finland, among other countries, has published the NAP on AMR for 2017–2021 [4]. The NAP highlights the importance of responsible antimicrobial use and sets the goal of decreasing antibiotic use. The favorable finding of decreasing antibiotic use in our study is in line with the goals of the NAP and likely due to several factors. First, the growing awareness of the AMR threat in Finland [6,8] is probably the most important factor. Secondly, pneumococcal as well as influenza vaccinations have also contributed to the decreased use of antimicrobial drugs [30,31]. The implementation of AMR action plans by the WHO [2] and EU [3], as well as our NAP in Finland [4], have likely also contributed to the favorable finding of the present study.

In this work, we found that, in addition to a 26% decrease in overall antibiotic use, this decrease was also found in all antibiotic categories, including wide-spectrum antibiotics, e.g., quinolones (decreased by 49%) and tetracyclines (decreased by 39%). This finding is also in line with the goals of the NAP in Finland [4]. It is noteworthy that the 10 most frequently used antibiotics covered 89% of all adult antibiotic prescriptions during the study period in Finland. Cefalexin was the most common antibiotic used during the study period; however, it showed a clear decreasing trend in the prescription rate. The decrease was also found in azithromycin and doxycycline. However, their use slightly increased from 2010–2011, which was also seen in our previous study among children and adolescents [9] during the same years, probably due to the large *Mycoplasma pneumoniae* epidemic observed in Finland from 2010–2011 [32].

Antibiotic use decreased in every age group during the study period. However, the antibiotic prescription rate was notably higher among those people over 80 years of age than in the other age groups throughout the whole study period, despite the overall decrease in antibiotic use in older adults. Specifically, the prescription rates of trimethoprim and pivmecillinam were considerably higher in the oldest adults compared to those of other age groups. Since these narrow-spectrum antibiotics (trimethoprim and pivmecillinam) are used only for treating cystitis, it can be concluded that the higher antibiotic prescription rate found among the over 80 years old most likely was due to older patients’ increased susceptibility to urinary tract infections [33]. In Sweden, there is a higher use of antibiotics for over 80-year-olds in nearly all of the main categories except macrolides [34]. In addition, the prescription rate for women was higher than for men in our study, thus confirming the results from previous research [11,13,17,18,20]. The gender difference in antibiotic prescribing may be partly explained by the higher prevalence of urinary tract infection antibiotic use in women [11,13] and also by the more active consultation behavior in women [35].

The overall annual costs of the use of outpatient antibiotics decreased but the antibiotic cost per prescription increased slightly. A similar phenomenon was found in our previous study concerning children and adolescents. This can be a result of, e.g., the increased price of antibiotics or increased use of more expensive antibiotics. According to both of our studies, antibiotic costs per prescription were highest in the elderly and children [9]. Although antibiotics are old products and included in generic substitution in Finland and, thus, are relatively cheap, they may cause an increased financial burden for pensioners and families with children who are already in a precarious financial situation, especially when they need the repeated use of antibiotics. It is also noteworthy that the total costs of cefalexin and amoxicillin/beta-lactamase inhibitors fluctuated strongly during the study period, likely due to price competition caused by the generic substitution and reference price system [24,36]. Interestingly, the arrival of a new competing cefalexin product to the market is a likely reason for the steep decrease in the total costs of cefalexin between 2012 and 2013 [36].

We have now found that outpatient antibiotic use, especially the use of broad-spectrum antibiotics, has decreased among the entire Finnish population. Overall, the results of our studies show that Finland has succeeded in reducing outpatient antibiotic use, which is in line with the AMR action plans [2,3,4]. However, it is noteworthy that, during the same time period, there was an increasing trend of ESBL *E. coli* strains in Finland, reflecting an increasing Gram-negative AMR health threat [7,8], as was apparent elsewhere in the world [1]. Based on our important findings in Finland, it can be concluded that controlling antibiotic use alone is insufficient to resolve the Gram-negative challenge. One likely explanation for increasing *E. coli* resistance, despite decreasing antibiotic use in Finland, is travel; studies by Kantele et al. from Finland have shown ESBL-carriers after traveling (especially) outside the EU area [37,38]. It can be concluded that increasing Gram-negative AMR, despite decreasing antibiotic use, calls for new innovations as soon as possible. These innovations could be a new class of antibacterial drugs with a new mechanism of action but also new anti-inflammatory innovations, such as nonantimicrobial, anti-inflammatory MMP (matrix metalloproteinase)-inhibitors or other new innovative therapies to block microbe–host interaction [39]. The efficacy and safety of new innovations have to be shown according to adequate regulatory rules in phase I, II, and III studies, which need both time and resources. These new innovations would probably have less of an AMR environmental risk than classical, old antibacterial drugs, for which regulatory challenges concerning ERA (Environmental Risk Assessment) have recently been discussed [40]. According to the recent joint report on EMA, ECDC, OECD, and EFSA [41], AMR is a silent pandemic, and this global health threat can only be tackled by increased coordination between the EU and worldwide.

The main strength of the present study is that it is nationwide, covering the reimbursed antibiotic purchases in outpatient care for the whole adult population in Finland. The other important strength is that the period covered by the study is fairly long, which provides the possibility of studying the long-term trends in the prevalence of antibacterial prescriptions and also cost change trends. However, there are also some limitations in this work. This work does not include nonreimbursable antibacterial drugs. However, the majority of antibacterial drugs used in adults in Finland have a reimbursement status. However, according to pharmaceutical sales, data recovered from the Finnish Medicines Agency Fimea for the total consumption of antibiotics in outpatient care showed a similar decreasing trend to that observed in our study, which indicates that the nonreimbursable drugs have minor relevance in terms of the reliability of the study. Furthermore, there are no data on the dosage or indication of the drugs, nor information on the actual use of the drugs after purchasing.

In conclusion, our findings show the decreasing trend of the use of antibacterial drugs in Finland among the whole outpatient population for one decade and are in line with AMR action plans. However, during the same time period, there has been an increasing trend of the Gram-negative AMR threat in Finland. Our important results from Finland further support the significance of the recent EMA, ECDC, EFSA, and OECD joint reports. According to the joint report, the global AMR health threat can only be tackled by increased coordination within the EU and worldwide by developing new innovative treatments, including antibacterial drugs with new mechanisms of action and using existing therapies responsibly around the world to ensure effective treatment for infectious diseases in the future. The new innovations are needed as soon as possible.

## Figures and Tables

**Figure 1 antibiotics-11-01453-f001:**
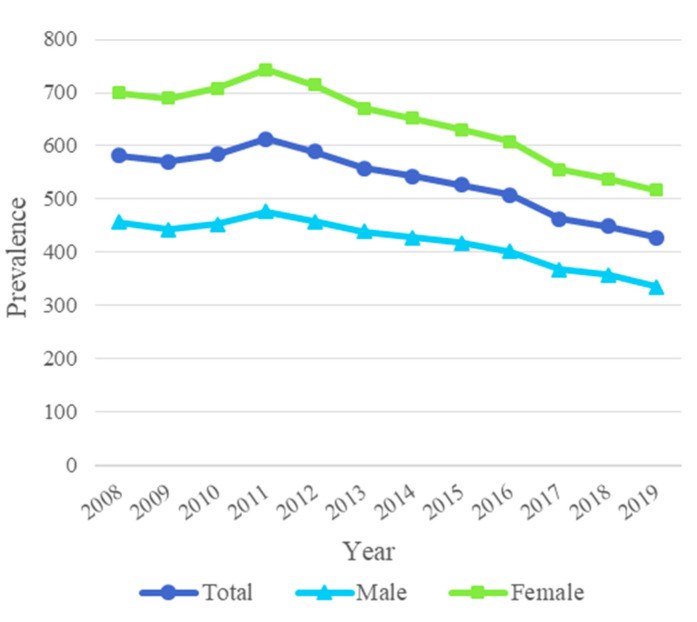
Decreasing trend in the antibiotic prescription rate per 1000 adults in Finland in 2008–2019.

**Figure 2 antibiotics-11-01453-f002:**
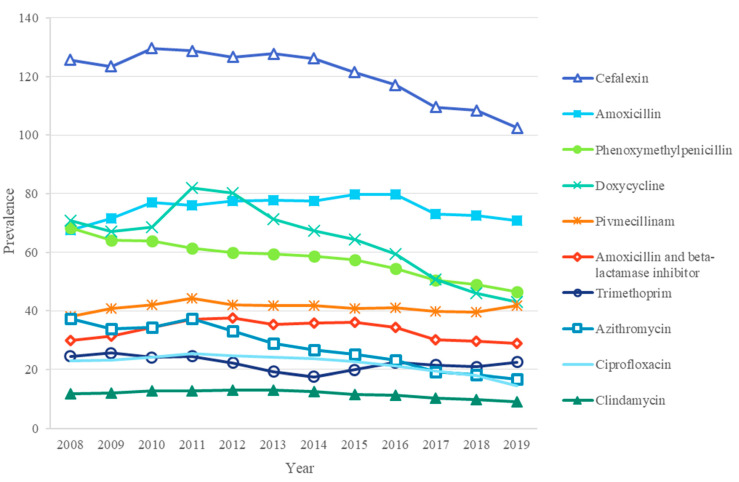
Prescription rate (prescriptions per 1000 adults per year) of the ten most frequently used antibiotics in Finland from 2008–2019.

**Figure 3 antibiotics-11-01453-f003:**
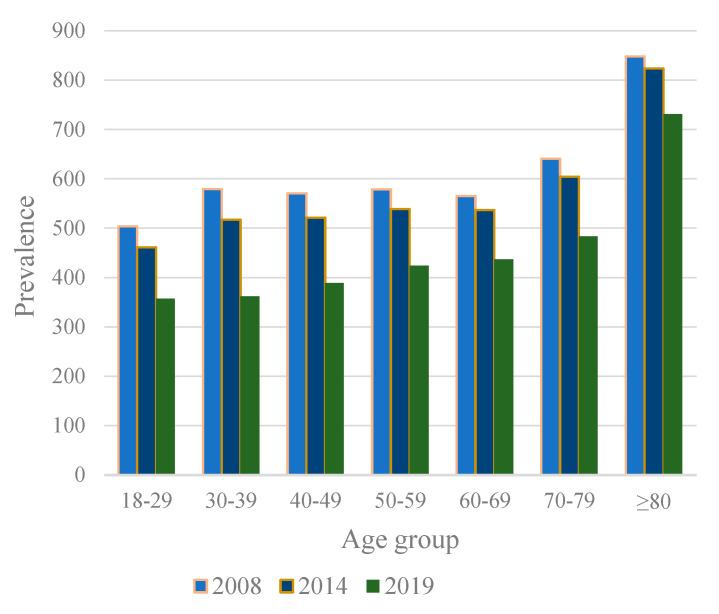
Antibiotic prescription rate per 1000 adults by age group at the beginning (2008), the middle (2014), and the end (2019) of the study period in Finland.

**Figure 4 antibiotics-11-01453-f004:**
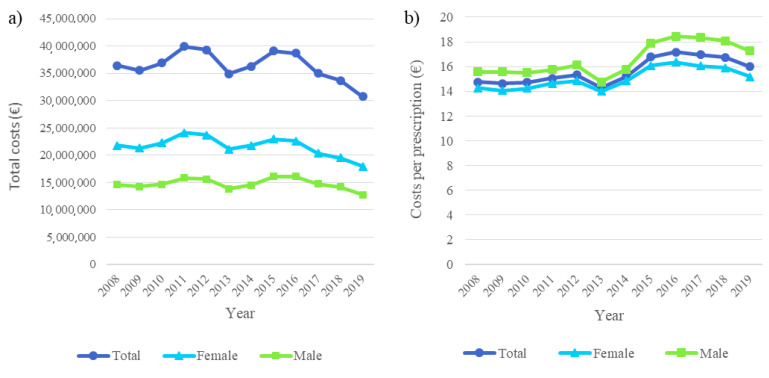
Antibiotic costs in Finland from 2008–2019. (**a**) Total costs. (**b**) Costs per prescription.

**Figure 5 antibiotics-11-01453-f005:**
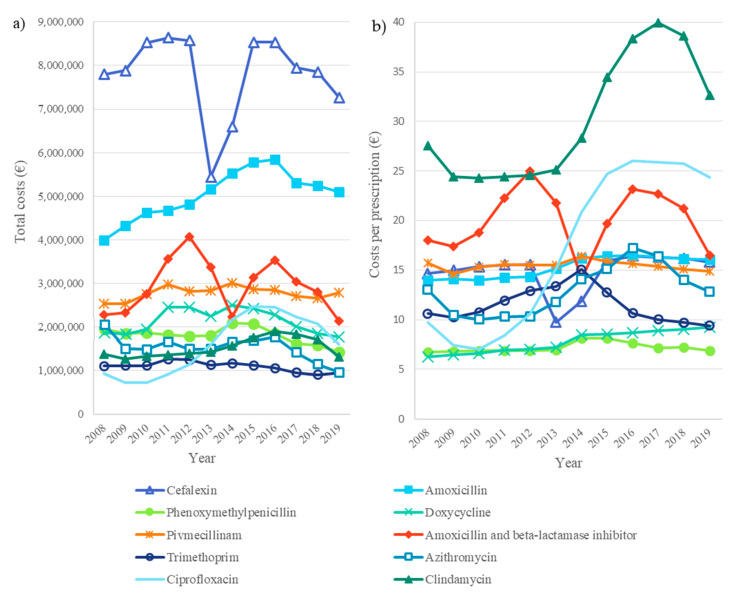
Trends of antibiotic costs for the 10 most frequently used antibiotics in Finland from 2008–2019. (**a**) Total costs. (**b**) Costs per prescription.

**Table 1 antibiotics-11-01453-t001:** Changes in antibiotic prescription rate per 1000 adults by antibiotic category (ATC-codes) between the years 2008 and 2019.

		Prescription Rate (Number of Prescriptions/1000 Adults/Year)		
Antibiotic Category	ATC Code	2008	2019	∆ (%)
Tetracyclines	J01A	85	52	−39%
Beta-lactam antibacterials, penicillins	J01C	207	195	−6%
Other beta-lactam antibacterials	J01D	133	104	−22%
Sulfonamides and trimethoprim	J01E	34	24	−29%
Macrolides, lincosamides and streptogramins	J01F	80	32	−60%
Quinolone antibacterials	J01M	43	22	−49%
All antibacterials for systemic use *	J01	582	428	−26%

* Aminoglycoside antibacterials (J01G) and other antibacterials (J01X) are not presented in the table because of the low number of prescriptions (<0.1 prescriptions/1000 adults/year). However, they are included in the total number of antibiotic prescriptions per 1000 adults per year.

## Data Availability

The statistical data used in this study are publicly available upon request from the Statistical Unit of the Social Insurance Institution, Finland.

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
