# Peer review of "Outpatient Antibiotic Use and Costs in Adults: A Nationwide Register-Based Study in Finland 2008–2019"

_antibiotics, 2022, doi:10.3390/antibiotics11111453_

Round 1

Reviewer 1 Report

The Finnish authors have performed a simple but potentially useful study showing the trend of outpatient antibiotics' prescription over a 12-year period.

Some points to be addressed follow.

1) The introduction is too long and needs to be shortened. The reference to E. coli and K. pneumoniae is perfunctory, since the preamble about the general problem of AMR worldwide suffices.

2) The same applies to discussion, that can be shortened, without too much reference to the rise of specific bacteria such as E. coli, since the focus of the paper is just outpatient antibiotics' prescription, without focussing on hospital consumptions of drugs.

3) The reference 41 is self-citation and is not essential.

Author Response

Response to Reviewer 1’s comments

We thank the reviewer for the kind comments.

The Finnish authors have performed a simple but potentially useful study showing the trend of outpatient antibiotics' prescription over a 12-year period.

Some points to be addressed follow.

1) The introduction is too long and needs to be shortened. The reference to E. coli and K. pneumoniae is perfunctory, since the preamble about the general problem of AMR worldwide suffices.

Response 1:

We have shortened the Introduction according to the referee’s comments: Instead of three paragraphs there is two paragraphs after the revision. The word count of the Introduction decreased from 1212 to 1035 words. However, we have not deleted the parts concerning E. coli and its’s AMR challenge with references, because we regard it as crucially important. This topic has been in focus in Finland since the beginning of early 2000’s. Noteworthy, in Finland this important gram negative health threat has also been a part of public discussion (described in the Finnish reference by Maarit Wuorela and Jari Jalava, the head of the expert group working on the local NAP). In the introduction we refer to this backround and regard it as the most important reason to set up this work.

2) The same applies to discussion, that can be shortened, without too much reference to the rise of specific bacteria such as E. coli, since the focus of the paper is just outpatient antibiotics' prescription, without focussing on hospital consumptions of drugs.

Response 2:

We thank for the comment, and have now shortened the discussion. After the revision the discussion part of the manuscript consists of 8 paragraphs instead of 10 paragraps, and the word count has decreased from 2034 words to 1728 words. However, when we planned this study, one important reason to carry out this study project was the increasing trend of E. coli ESBL strains in the outpatient wards, as Wuorela and Jalava in the Finnish publication point out. That´s why we focused this study on outpatient antibiotic consumption. In Finland over 90% of  total antibiotic consumption is outpatient consumption. Additionally, we want to point out that a recent publication this year concerning the global AMR burden indicates that gram negative threat is a very important global health threat. Please, see reference Antimicrobial Resistance Collaborators. Global burden of bacterial antimicrobial resistance in 2019: a systematic analysis. Lancet 2022; 399: 629–655. doi: 10.1016/S0140-6736(21)02724-and especially Figure 4 in the reference showing that E. coli and K. pneumonea as gram negative bacteria are major global challenges.

3) The reference 41 is self-citation and is not essential.

Response 3:

Reference 41 is mentioned as an example of self-citation. We do agree, that it is thus not essential. However,  according to our opinion self-citation is not denied in the scientific discussion as part  of the article. That’s why the decided to keep this citation, in order to emphasize that for many decades there haven’t been any new antibiotic innovations. We want to encourage to find new  solutions to resolve increasing gram negative health  threat, which we regard as a very serious health threat, a silent pandemic  (reference: ECDC, EMA, OECD, IFSA Joint report date 7.3.2022, Lancet 2022). Furthermore, a new sentence concerning the need for adequate regulatory documentation is added for clarification on rows 555-557 in the manuscript: The efficacy and safety of new innovations have to beshown according to adequate regulatory rules in phase I, II and III studies, which need both time and resources..”

Reviewer 2 Report

The article describes the use and costs of antibiotics in the outpatient setting by age and sex among adults in Finland during 2008-2019. I found it well documented and complete, and have only some minor comments:

-     -  The prevalence seems to increase until 2011 followed by a decrease of use afterwards as you mentioned in l. 120 "The antibiotic prescription rate was highest in 2011 (614 prescriptions per 1000 adults), after which 121 the rate decreased steadily.". Can you explain this change by an implementation of antibiotic stewardship?  or is the higher use during this period only due to prescriptions made during the Mycoplasma pneumoniae epidemic?

-    -   Not much publications explore the differences of antibiotic use by sex.Please could you discuss it a bit more in details ? Have you some publications of antibiotic prescriptions for simple cystitis in Finland which could support your findings (too much prescriptions for this indication f.e.) ? Or even without having the indications would it be possible to draw conclusions for this difference?

-      - Figure 3:  I would suggest to replace the line chart by a bar chart (3 years for each age category f.e.).

-   - Could it be possible to add statistics to support your findings, f.e. for trend between 2008 and 20019 for use and costs ?

-     -  If possible, I would delete the references in Finnish.

Author Response

Response to Reviewer 2’s comments

We thank the reviewer for the kind comments.

The article describes the use and costs of antibiotics in the outpatient setting by age and sex among adults in Finland during 2008-2019. I found it well documented and complete, and have only some minor comments:

1) -     -  The prevalence seems to increase until 2011 followed by a decrease of use afterwards as you mentioned in l. 120 "The antibiotic prescription rate was highest in 2011 (614 prescriptions per 1000 adults), after which 121 the rate decreased steadily.". Can you explain this change by an implementation of antibiotic stewardship?  or is the higher use during this period only due to prescriptions made during the Mycoplasma pneumoniae epidemic?

Response 1:

The decrease of the outpatient antibiotic use is most likely due to increased awareness of antimicrobial resistance and long-term and subtle antibiotic stewardship in the national guideline work (Current Care guidelines). For example the Current Care guidelines on UTI, sexually transmitted diseases, acute laryngitis, acute dental infections and lower respiratory tract infections have been constantly updated during our study time. However, it is difficult to track the old versions of the Current Care guidelines and due to their multiple changes also the effect on antibiotic use, and therefore we have not referred to them in the text. There is an English page for the Current Care guidelines (https://www.kaypahoito.fi/en/guidelines), but it only includes parts of five of the 102 guidelines currently in place. There has not been any large scale antibiotic stewardship program between the Mikstra program (https://pubmed.ncbi.nlm.nih.gov/16586379/) in the 1990’s and National Action Plan (2017-2022).   

The reviewer acknowledges the Mycoplasma pneumonia epidemic, which truly seems to have affected the total consumption of antibiotics in 2011. Please, see also the sales figures in the Response 4.

2) -    -   Not much publications explore the differences of antibiotic use by sex. Please could you discuss it a bit more in details ? Have you some publications of antibiotic prescriptions for simple cystitis in Finland which could support your findings (too much prescriptions for this indication f.e.) ? Or even without having the indications would it be possible to draw conclusions for this difference?

Response 2:

We thank the reviewer for an important comment. We had indeed no discussion on the sex difference in the antibiotic use in our article. Now we have added a short comparison to previous studies into the discussion on rows xxx-xx. Unfortunately, we did not find any recent studies for prescribing or medicine use in cystitis from Finland.

Rows 271-275: “In addition, the prescribing rate for women was higher than for men in our study, thus confirming the results from previous research 11, 13, 17, 18, 20. The gender difference in antibiotic prescribing may be partly explained by the higher prevalence of urinary tract infection antibiotic use in women 11, 13, and also by the more active consultation behavior in women 35.”

3)     - Figure 3:  I would suggest to replace the line chart by a bar chart (3 years for each age category f.e.).

Response 3:

Thank you for a good suggestion. We have now replaced the line chart by a bar chart.

The chart can also be found in the updated excel file with all the figures.

-4)    - Could it be possible to add statistics to support your findings, f.e. for trend between 2008 and 20019 for use and costs ?

Response 4:

The sales statistics for total population in outpatient and instutional care in DDD/1000 inhabitants/day indeed confirm our results, see the figure and table below. We have mentioned this on the rows 555-558 in the strengths and limitations paragraph. However, the sales statistics figures are only available for the total population, and cannot be produced for the adult population like our study data. Also distinguishing the outpatient sales from hospital sales is impossible in the given time frame (5 days including a weekend). Therefore, we considered, that adding these figures as supplementary table would potentially confuse the readers. Despite the difference in the populations the shape of the curves in figure 1 and the figure presented below, are similar, including the peak in consumption in 2011. Due to the problem of not being able to provide sales figures for adult outpatient population, we did not produce the table and figure in terms of euros, but just used the DDD/1000 inh/day consumption figures to provide an example.

2006

2007

2008

2009

2010

2011

2012

2013

2014

2015

2016

2017

2018

2019

J01 DDD/

1000 inh/

day

22.12

22.73

23.1

23.44

23.32

24.63

22.00

20.90

20.66

19.74

19.04

17.33

15.34

14.69

5)     -  If possible, I would delete the references in Finnish.

Response 5:

We do understand the difficulty with the Finnish language. However, in our opinion it is very important to present this reference in this context. The reference is an editorial in a leading medical journal in Finland and it is written by leading national experts dealing e.g. with the NAP. This reference reflects the public discussion in Finland concerning challenges of the outpatient therapy of resistant E. coli induced urinary tract infection. Therefore, we think that this reference is important in this context to clarify the important role of E. coli AMR resistance.  
